# Impact of Subsyndromal Delirium Occurrence and Its Trajectory during ICU Stay

**DOI:** 10.3390/jcm11226797

**Published:** 2022-11-17

**Authors:** Rodrigo B. Serafim, Felipe Dal-Pizzol, Vicente Souza-Dantas, Marcio Soares, Fernando A. Bozza, Pedro Póvoa, Ronir Raggio Luiz, José R. Lapa e Silva, Jorge I. F. Salluh

**Affiliations:** 1Instituto D’Or de Pesquisa e Ensino, Rio de Janeiro 22281-100, Brazil; 2Hospital Copa D’Or, Rio de Janeiro 22031-011, Brazil; 3Hospital Universitário Clementino Fraga Filho/Instituto de Doenças do Tórax, Universidade Federal do Rio de Janeiro, Rio de Janeiro 21941-913, Brazil; 4Laboratório de Fisiopatologia experimental, Programa de Pós-Graduação em Ciências da Saúde, Universidade do Extremo Sul Catarinense, Criciúma 88806-000, Brazil; 5Programa de Pós-Graduação em Clínica Médica, Universidade Federal do Rio de Janeiro, Rio de Janeiro 21941-913, Brazil; 6Instituto de Pesquisa Clínica Evandro Chagas, FIOCRUZ, Rio de Janeiro 22281-100, Brazil; 7Unidade de Cuidados Intensivos Polivalente, Hospital de São Francisco Xavier, Centro Hospitalar de Lisboa Ocidental, 1150-199 Lisboa, Portugal; 8NOVA Medical School, CEDOC, Universidade Nova de Lisboa, 1150-082 Lisboa, Portugal; 9Center for Clinical Epidemiology and Research Unit of Clinical Epidemiology, OUH Odense University Hospital, C 5000 Odense, Denmark; 10Instituto de Saúde Coletiva, Universidade Federal do Rio de Janeiro, Rio de Janeiro 21941-592, Brazil

**Keywords:** subsyndromal delirium, delirium, ICU, critically ill, cognitive dysfunction

## Abstract

Despite recent advances in the field, the association between subsyndromal delirium (SSD) in the ICU and poor outcomes is not entirely clear. We performed a retrospective multicentric observational study analyzing mental status during the first 72 h of ICU stay. Of the 681 patients included, SSD occurred in 22.7%. Considering the worst cognitive assessment during the first 72 h, 233 (34%) patients had normal mental status, 124 (18%) patients had SSD and 324 (48%) patients had delirium or coma. SSD was not independently associated with an increased risk of death when compared with normal mental status (OR 95%IC 1.0 vs. 1.35 [0.73–1.49], *p =* 0.340), but was associated with a longer ICU LOS (7.0 (4–12) vs. 4 (3–8) days, *p <* 0.001). SSD patients who deteriorated to delirium or coma (21%) had a longer ICU LOS in comparison with those who improved or maintained mental status (8 (5–11) vs. 6 (4–8) days, *p =* 0.025), but did not have an increase in mortality. The main factors associated with the progression from SSD to delirium or coma were the use of mechanical ventilation, the use of intravenous benzodiazepines and a baseline APACHE II score > 23 points. Our findings support the association of SSD with increased ICU LOS, but not with ICU mortality. Monitoring the trajectory of SSD early at ICU admission can help to identify patients with increased risk of conversion from SSD to delirium or coma.

## 1. Introduction

Despite the perception that subsyndromal delirium (SSD) is associated with worse outcomes of intensive care unit (ICU) patients [1,2], current evidence is derived mostly from studies with small sample sizes or heterogeneous ICU populations [1,2,3,4]. This may partly explain the contradictory results regarding the impact of SSD on clinical outcomes such as mortality and length of stay (LOS) [5,6]. Studies of non-ICU patients demonstrate that SSD increases hospital LOS, post-discharge mortality and functional decline [7,8]. However, a meta-analysis evaluating exclusively ICU patients concluded that SSD was not associated with increased risk of death, although its occurrence could be associated with increased hospital LOS [9].

Regarding ICU delirium, several studies demonstrated that the duration of delirium and its severity are the key factors associated with worse outcomes [10,11,12]. In fact, a very short duration of delirium seems to have little impact on the mortality rates of ICU patients [13,14]. However, it remains unclear for SSD. In a recent publication, after adjusting for covariates, the duration of SSD was an independent predictor of post-hospital institutionalization [10]. Moreover, the cognitive trajectories of patients presenting with SSD and the impact on outcomes are not entirely clear.

In clinical practice, multiple scenarios are possible and patients with SSD can evolve back to normal cognition or deteriorate to delirium or coma during ICU stay [15,16]; the later conditions are known to be associated with worse outcomes [17]. Therefore, it is reasonable to hypothesize that the trajectory of SSD should influence outcomes as it could be considered to be a worsening of cognition in previously “normal” subjects, or an intermediate phase (of resolution) of cognitive impairment when patients presenting with delirium or coma evolve to SSD before full recovery.

Two studies have evaluated the transition from SSD to delirium and described the effect of antipsychotic drugs in preventing conversion from SSD to delirium with conflicting results [15,16]. Hakim et al. described a significantly lower incidence of delirium with the administration of risperidone when compared with placebo (13.7% vs. 34%, *p* = 0.031) [15], and Al Qadheeb et al. only described a reduction in the duration of agitation with the use of haloperidol [16]. Neither of these studies clearly demonstrated any other impact in relevant outcomes, such as ICU and hospital LOS or mortality.

To address these knowledge gaps, we performed a retrospective multicentric cohort study to describe the impact of SSD occurrence and its trajectories on ICU outcomes. Moreover, we evaluated the factors potentially associated with progression from SSD to delirium or coma.

## 2. Materials and Methods

This cohort study was conducted at the intensive care units of three tertiary hospitals in Brazil (Hospital Copa D’or, Rio de Janeiro, RJ; Instituto Nacional do Câncer, Rio de Janeiro, RJ; Hospital São José, Criciúma, SC). The Institutional Review Boards at each institution approved the study (approval numbers, respectively: 14/05; 144/2009; 664.794). Consecutive adult patients were selected and only those admitted for >48 h to the medical and surgical ICU at each institution were included.

The Confusion Assessment Method for the Intensive Care Unit (CAM-ICU) validated for the Brazilian population was used daily to assess for SSD in patients [18]. CAM-ICU was applied for nurses (in Hospital Copa D’or, Rio de Janeiro, RJ and Hospital São José, Criciúma, SC, Brazil) and for respiratory therapists and nurses (Instituto Nacional do Câncer, Rio de Janeiro, RJ, Brazil). Before the study started, health professionals involved in data collection were trained to systematically diagnose delirium. The CAM-ICU training program contained information about delirium, including the relevant literature, handouts, and a video about the detailed application of the CAM-ICU. The diagnosis of delirium in CAM-ICU was compared with the diagnosis made by a team of psychiatrists and neurologists with expertise in delirium. The data collected were compared at the same time of day to ensure consistency between observers. Delirium was considered present if the patient demonstrated an acute change in mental status or a fluctuating course of mental status and inattention, plus either altered consciousness or disorganized thinking [19,20]. SSD was considered present if the assessment was negative, but the patient exhibited any CAM-ICU features [8]. Patients who could not be assessed for delirium at any time during the study period (patients with Richmond Agitation–Sedation Scale scores (RASS) [16] of −4 or −5 during the study period), under 18 years old, patients expected to die <24 h and patients admitted as a result of brain trauma or other primary neurological reason were excluded.

Considering that delirium occurrence is higher in the first days after ICU admission [21,22] and up to 70% of patients may present delirium at admission [14], mental status during the first three days in ICU was of special interest to the analysis. The objective was to evaluate the outcomes (particularly ICU LOS and mortality) based on the type of early brain dysfunction (delirium and SSD) and its early trajectory.

### 2.1. Data Collection and Definitions

Patients were screened once daily using the CAM-ICU. The mental status in the first three days after admission was considered for analysis. The patients were followed until hospital discharge. Demographic data, surgical and medical diagnoses, comorbidities, laboratory results and the use of benzodiazepines during the period were registered in a structured case report form. Acute Physiology and Chronic Health Evaluation (APACHE) II scores were calculated using data from the first 24 h of the ICU admission [23].

With the results of the CAM-ICU in the first 72 h of admission, patients were classified according to the worst mental status in the period. This classified patients initially in 3 groups: (1) normal status (the CAM-ICU assessment was always negative in all features), (2) SSD (the CAM-ICU assessment was negative for delirium, but the patient demonstrated at least 1 CAM-ICU feature) and (3) delirium or coma (the CAM-ICU assessment was positive or RASS of −4 or −5 during the study period). Subsequently, we analyzed SSD trajectories and all patients that presented SSD as a first form of cognitive dysfunction were also classified according to the cognitive trajectory: (1) improvement to normal mental status, (2) sustained SSD status or (3) progression to delirium or coma.

### 2.2. Statistical Analysis

In our cohort study, continuous variables are summarized as medians and interquartile ranges (IQR = 25th and 75th percentiles), and dichotomous categorical variables are presented as proportions of frequency. Nonparametric chi-squared tests were performed to compare the association between the clinical and demographic variables and the development of SSD or delirium or coma. The main outcomes of interest were ICU LOS and mortality of any cause. A *p* value < 0.05 was considered significant. After univariate analysis, variables that presented a *p* < 0.25 were entered in the multivariate analysis to correlate the worse mental status with the risk of death.

We performed multivariate forward logistic regression relating the results of APACHE II score higher than 23 (84th percentile), surgical or medical conditions, use of benzodiazepines, age (>65 yo) and the presence of SSD or delirium or coma with mortality. Multivariate logistic regression was also used to identify the factors associated with the progression of SSD to delirium or coma.

## 3. Results

After the initial screening of 878 ICU patients, 681 were included in this study. Considering the worst cognitive assessment in the first 72 h, 233 (34%) patients had normal mental status, 124 (18%) patients had SSD and 324 (48%) patients had delirium or coma. Patients with delirium or coma had a higher baseline severity of illness (as expressed by a higher APACHE II score), used more invasive mechanical ventilation and were more frequently admitted due to medical reasons compared to those with normal cognition or SSD (Table 1).

The median (IQR) lengths of stay for each of the worse mental status groups, compared to normal mental status, SSD, and delirium or coma groups were, respectively, 4 (3–8), 7.0 (4–12), and 15 (9–24) days, with normal vs. SSD, *p* = 0.035; SSD vs. delirium or coma, *p* = 0.014 and normal vs. delirium or coma, *p* < 0.001 (Figure 1). Mortality rates were, respectively, 21%, 30.6% and 47.5%, with normal vs. SSD *p* = 0.035, SSD vs. delirium or coma *p* < 0.013 and normal vs. delirium or coma *p* < 0.001. In a multivariate analysis comparing the normal mental status, SSD and delirium or coma groups, they were not independently associated with increased risk of death, respectively, with 1.0, 1.35 (0.73–1.49) and 1.09 (0.56–1.88), *p* = 0.594. To evaluate the SSD impact in low-severity patients, we separated our population according to the median APACHE II score (12 points). The group with SSD and APACHE II < 12 points presented a trend towards increased mortality, the ORs, respectively, with 1.0, 2.91 (0.95–8.92) and 3.00 (1.11–8.14), and normal vs. SSD, *p* = 0.061 and normal vs. delirium or coma, *p* < 0.031.

At admission, 103 (15%) patients had SSD. Additionally, 44 (6.4%) patients on the second day and 8 (1.2%) patients on the third day progressed from normal to SSD. Patients who only developed SSD on the third day of ICU stay were not considered for the analysis of cognitive trajectory. Of the 147 patients who presented SSD as the first form of cognitive dysfunction, 67 (45.6%) improved to normal mental status, 49 (33.3%) remained in SSD status and 31 (21%) progressed to delirium or coma.

We compared the group of SSD patients who improved or maintained mental status with the group who progressed to overt delirium or coma. The median LOS (IQR) was, respectively, 6 (4–8) days, 8 (5–11) days, *p* = 0.025. The crude mortality rates were, respectively, 20.7% and 35.9%, *p* = 0.034. However, when multivariate analysis was performed, we could not observe an independent association of SSD trajectory with mortality, respectively, 1.32 (0.83–9.27) and 1.38 (0.56–20.73), *p* = 0.18) (Appendix A).

The use of mechanical ventilation was the main condition presenting a strong association with progression from SSD to delirium or coma (OR = 10.98 (6.0–20.0), *p* < 0.001). Among other factors potentially associated with progression from SSD to delirium or coma, we analyzed the use of intravenous benzodiazepines (OR = 3.0 (0.8–11.2), *p* = 0.156) and an APACHE II score higher than 23 (84th percentile) (OR = 3.98 (0.9–12.0), *p* = 0.078) (Table 2).

Finally, 93.2% of the patients with SSD presenting with the three conditions (need for MV, sedation with benzodiazepines and an APACHE II score higher than 23) evolved to delirium or coma. We analyzed the individual features of the CAM-ICU, but It was unable to identify any CAM-ICU features associated with the progression from SSD to delirium or coma (Appendix A).

## 4. Discussion

The present cohort study evaluated the impact of SSD and its different trajectories on clinical outcomes of critically ill patients. Our findings demonstrate that the prevalence of early SSD in ICU patients is 22.7%, confirming the notion that it is a prevalent condition. Moreover, 103 (15%) patients at day one and an additional 7% in the following 2 days presented SSD, which was the only sign of early mental dysfunction in the ICU for these patients.

Interestingly, in our study, overall SSD was not associated with higher mortality compared to patients without cognitive impairment. Nonetheless, when considering patients with a lower severity of illness (APACHE II < 12), the occurrence of SSD showed a trend towards increased mortality (*p* = 0.064). This finding is in accordance with the literature that shows that SSD has a significant clinical impact in hospitalized patients with lower severity that are not critically ill [8]. The occurrence of SSD (a condition of lower severity compared with rapidly reversible delirium) may not be sufficient to generate worse outcomes in the ICU. The SSD may be a marker for underlying medical conditions not severe enough to cause full delirium in non-ICU population where the cognitive trajectory and baseline severity of illness leads to a slowly increasing number of risk factors. In contrast, in the ICU, where many risk factors are usually present in the first day and often occur simultaneously, the patient converts rapidly to delirium or coma.

In addition, we also show that SSD can lead to different outcomes depending mostly on the baseline severity of illness and the cognitive trajectories of patients. According to our findings, in SSD, a worsening early cognitive trajectory is associated with increased LOS. Mechanical ventilation was the main factor associated with the progression from SSD to delirium or coma (Table 2).

The present study also shows that, although findings of abnormal cognition in a single and baseline assessment may be associated with outcomes, the trajectory of SSD is informative to understand its full impact on outcomes. Our data show that SSD status is often a transitory stage that evolves quickly with either normalization or worsening to delirium or coma, this later finding is especially frequent in mechanically ventilated patients.

In our cohort, we could not identify any CAM-ICU feature associated with the progression from SSD to delirium or coma. Although inattention has been described as a cardinal feature of the full delirium syndrome [19,24] it was not observed in SSD [10] and a full screening of the features should be always considered.

The present study has some limitations. First, although our sample was one of the largest, the number of patients enrolled was relatively limited for subgroup analysis. This limits our potential to evaluate the prognostic value of each feature of the CAM-ICU and reduces the power to associate the variables described with progression from SSD to delirium or coma. Second, we only described the trajectory of SSD in the first 3 days in ICU, but this may represent a bias specially to long-stay patients. Third, delirium was screened once a day, which can contribute to a lower diagnosis especially of SSD. Fourth, we did not have an electroencephalogram or other type of continuous brain monitoring. Heavily sedated patients were excluded, and some hypoactive delirium could be missed. Finally, the evaluation of patients with the CAM-ICU dichotomizes the diagnosis of mental status. The use of quantitative scales such as the ICDSC [25] and the new delirium-rating scales such as the CAM-S [26] and the CAM-ICU-7 [27] may be more suitable with the proposed diagnosis of SSD.

## 5. Conclusions

SSD is a prevalent condition and occurred in 22.7% of patients in our cohort. Our findings support the association of SSD with longer ICU LOS but not with ICU mortality. Monitoring the trajectory of SSD early at ICU admission can help to identify patients with worse prognosis (patients who deteriorate from SSD to delirium or coma). The use of mechanical ventilation, the severity of the critical illness and the use of benzodiazepine were the factors associated with progression from SSD to delirium or coma.

## Figures and Tables

**Figure 1 jcm-11-06797-f001:**
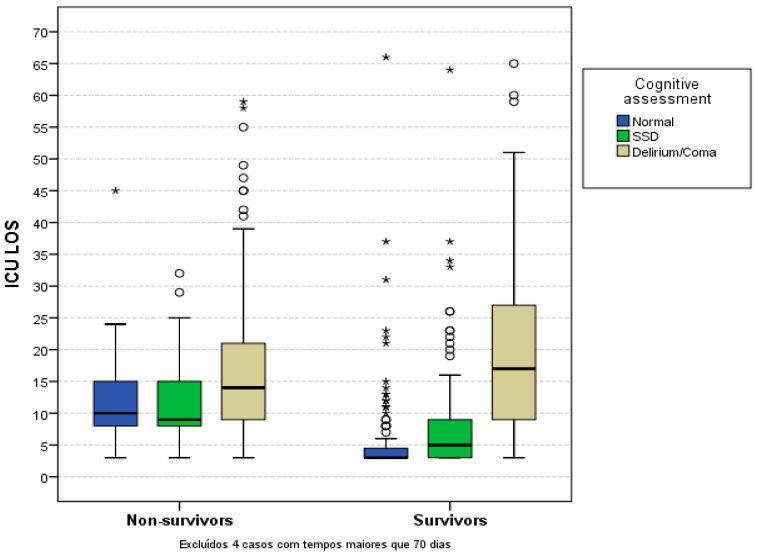
ICU LOS considering worse diagnosis in the first 72 h. Legend: The boxplot represents the median (IQR) of survivors and non-survivors according to mental status: normal, SSD and delirium or coma. The median (IQR) ICU LOS of non-survivors was, respectively, 10 (8–15), 9 (8–15), 14 (9–21) days, with normal vs. SSD *p* = 0.973, SSD vs. delirium/coma *p* = 0.014, normal vs. delirium/coma *p* < 0.001 and the survivors was, respectively, 3 (3–4.5), 5 (3–9), 17 (9–27) days, with normal vs. SSD *p* < 0.001, SSD vs. delirium/coma, *p* < 0.001, normal vs. delirium/coma *p* < 0.001. Four patients with more than 70 days of LOS were excluded. Each circle means: outlier patients and the asterisk means: extremely outlier patient.

**Table 1 jcm-11-06797-t001:** Demographic and clinical variables of patients according to the worse mental status in the first 72 h.

Variables	Total (*n* = 681)	Normal (*n* = 233)	SSD (*n* = 124)	Delirium/Coma (*n* = 324)	*p*-Value
Age (years)	61 (52–72)	60 (52–72)	61 (50–74)	62 (52–71)	0.242
Male gender, n (%)	378 (55.5%)	116 (50%)	61 (49.5%)	191 (58.8%)	0.880
Apache II score	12 (8–18)	11 (7–15)	11 (8–16)	15 (10–22)	<0.001
Benzodiazepin use	217 (31.9%)	41 (17.7%)	14 (11.3%)	162 (50%)	0.062
Medical, n (%)	270 (39.6)	42 (18.2%)	41 (33.0%)	187 (57.7%)	<0.001
Mechanical Ventilation, n (%)	306 (44.9%)	27 (11.36)	6 (4.8%)	273 (84.2%)	<0.001
Main diagnostic categories, n (%)					
Abdominal	227 (33%)	81 (34.7%)	38 (30.6%)	72 (22%)	
Pulmonary	143 (21%)	10 (4.3%)	37 (33%)	76 (23.5%)	
Cardiovascular	74 (10.9%)	62 (26.6%)	41 (32.8%)	77 (23.8%)	
Neurological (non-SNC)	28 (4.1%)	5 (2%)	12 (9.6%)	2 (0.6%)	
Genitourinary	34 (5%)	13 (5.8%)	10 (8.1%)	10 (3%)	
Trauma	11 (1.6%)	6 (2.6%)	3 (2,4%)	2 (0.6%)	
Orthopedic	59 (8.7%)	27 (11.6%)	15 (12.1%)	15 (4.6%)	
Duration of MV (days)	7.0 (3–14)	3.0 (2–6)	4.5 (2–8)	8.0 (4–16)	<0.001
Mortality, n (%)	241 (35.4%)	49 (21%)	38 (30.6%)	154 (47.5%)	0.03
LOS (days)	8.0 (4–16)	4.0 (3–8)	7.0 (4–12)	15.0 (9–24)	<0.001

Legend: continuous variables are summarized as medians and interquartile ranges (IQR = 25th and 75th percentiles).

**Table 2 jcm-11-06797-t002:** Risk factors to conversion from SSD to delirium/coma.

	OR (IC)	*p*-Value
Mechanical Ventilation	10.98 (6–20)	0.001
Benzodiazepines	3 (0.8–11.2)	0.29
Apache II > 23	3.98 (0.9–12)	0.065
Age > 65 yo	1.3 (0.6–2.8)	0.69
Surgical admision	1.9 (0.6–6)	0.168

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
