# Peer review of "Impact of Subsyndromal Delirium Occurrence and Its Trajectory during ICU Stay"

_jcm, 2022, doi:10.3390/jcm11226797_

Round 1
Reviewer 1 Report
The authors present their results about the impact of subsyndromal delirium in the ICU with a retrospective study of 681 patients. They found that although the ssd has an impact on ICU LOS, it does not alter the incidence of ICU mortality. I would like to congratulate the authors for this interesting study.
1) Line 80 there is a missing space
2) Based on guidelines the screening for delirium should be made every 8 hours in order to counteract for it's fluctuating course during the day. Hence, it is important to clarify if this recommendation was applied in your study sample.
Author Response
We want to thank you for having considered our paper for publication in JCM. We answered point-by-point the comments and highlighted all changes in the manuscript (in red font). We believe the manuscript is now much improved thanks to the reviewers’.
The authors present their results about the impact of subsyndromal delirium in the ICU with a retrospective study of 681 patients. They found that although the ssd has an impact on ICU LOS, it does not alter the incidence of ICU mortality. I would like to congratulate the authors for this interesting study.
1) Line 80 there is a missing space
Response: Thanks, it was corrected in the revised manuscript.
2) Based on guidelines the screening for delirium should be made every 8 hours in order to counteract its fluctuating course during the day. Hence, it is important to clarify if this recommendation was applied in your study sample.
Response: the reviewer is right to point this out. Once the mental status can fluctuate during the day, the number of formal assessments using CAM-ICU can contribute to a better detection of mental status deterioration. Unfortunately, our staff trained to use the CAM-ICU could not perform the evaluation more than once a day. It is a limitation described in many studies who evaluated delirium in ICU, but the assessment once a day is more reliable, and it reflects a real-life condition. We included this discussion as a limitation of the study.
Reviewer 2 Report
Major comments
1. There is a discrepancy between the objectives and the analysis. Specifically, the introduction describes the impact of SSD on outcomes, but this does not describe the impact of trajectory on outcomes.
2. Also, the analysis attempts to identify risk factors with trajectory as an outcome, but these are not included in the objectives of this study.
3. There needs to be some organization regarding the purpose of the study and the analysis to achieve the objectives.
4. The discussion is not organized. The discussion needs to be further organized.
5. This is an observational study and should not use terms that assume a causal relationship.
Minor comments
1. Line 116-117 The results here seem to be in Line 143-145, but it would be good to show the details in a table (should not as appendix).
2. Line 115-116 In the explanatory model, I think variables should be based on medical rationale and past findings.
3. Line 117 I think logistic regression is inappropriate when ICU LOS is used as an outcome. You need to describe in detail what type of analysis was performed.
4. Line 117 and Figure 1 If death is the outcome, I think Kaplan-Meier curves and Cox proportional hazards models are good indications, is there any reason why they were not done?
5. Table 2 APACHE II and age are binary variables, but this is not stated in the method.
6. Line 167 The authors seem to be trying to identify the factors that make SSD to delirium transition as an outcome. This does not appear to be consistent with the purpose of this study.
7. Line 180 The authors might want to use “associated with”, instead of “impact”.
8. Line 190 Where Table 3 is going to ?
9. We do not think it is necessary to include OR=1 for reference in the multivariate analysis table.
Author Response
We want to thank you for having considered our paper for publication in JCM. We answered point-by-point the comments and highlighted all changes in the manuscript (in red font). We believe the manuscript is now much improved thanks to the reviewers’.
Major comments
- There is a discrepancy between the objectives and the analysis. Specifically, the introduction describes the impact of SSD on outcomes, but this does not describe the impact of trajectory on outcomes.
Response: the reviewer is right to point this out. However, the literature about SSD trajectory is very limited and the introduction section was focused on delirium and SSD outcomes. We revised the introduction and discussion sections to include available data about the SSD trajectory and to clarify the objectives of the study. The text included were highlighted.
- Also, the analysis attempts to identify risk factors with trajectory as an outcome, but these are not included in the objectives of this study.
Response: We agree with the reviewer. We made changes in the introduction to clarify the objectives.
- There needs to be some organization regarding the purpose of the study and the analysis to achieve the objectives.
Response: Thanks for the commentaries. We made changes in the manuscript to clarify the objectives. We revised the manuscript to organize the objectives in 3 steps: (1) SSD prevalence and mental status impact in LOS and mortality (2) SSD trajectory and its impact in LOS and mortality (3) factors related with conversion of SSD to delirium or coma. Objectives and results are presented accordingly with these steps.
- The discussion is not organized. The discussion needs to be further organized.
Response: Thanks for the commentaries. We revised the manuscript and especially the discussion section. We included new comments on discussion, and we better organized sections accordingly with the results.
- This is an observational study and should not use terms that assume a causal relationship.
Response: Thanks for the commentaries. We revised the manuscript and made corrections in the text
Minor comments
- Line 116-117 The results here seem to be in Line 143-145, but it would be good to show the details in a table (should not as appendix).
Response: Thanks for the commentaries. We revised the manuscript and made corrections in the text
- Line 115-116 In the explanatory model, I think variables should be based on medical rationale and past findings.
Response: Thanks for the commentaries. We agree with the reviewer about the inclusion of variables accordingly with medical rationale. But all variables’ testes were potentially associated with outcomes based on clinical reasoning. So, we choose the model with those variables with a high significance based on the p-value <0,25. We believe that not assuming the traditional cut-off levels such as 0.05 we can successfully identify variables known to be important.
- Line 117 I think logistic regression is inappropriate when ICU LOS is used as an outcome. You need to describe in detail what type of analysis was performed.
Response: Thanks for the commentaries. We agree with the reviewer. The text was confusing. We edited the text to clarify that the multivariable analysis and logistic regression was used to correlate variables with mortality. LOS was compared using nonparametric tests.
- Line 117 and Figure 1 If death is the outcome, I think Kaplan-Meier curves and Cox proportional hazards models are good indications, is there any reason why they were not done?
Response: The reviewer is right to point this out. We performed the cox-regression for survival analysis, but it did not add any new information, so we decided to not include this result. We considered that due the small sample size in our study (considering only the SSD group) and the small follow-up time the cox-regression was not so relevant. We included below the cox-regression model for survival.
Characteristics |
Cox model univariable |
|
Cox model multivariable |
||
HR |
p-value |
|
HRaj* |
p-value |
|
SSD classification |
0,252 |
0,137 |
|||
Improveing |
1 |
1 |
|||
Stable |
1,10 |
0,107 |
0,89 |
0,161 |
|
Worsening |
2,00 |
0,818 |
|
1,68 |
0,419 |
Mechanical ventilation |
|||||
No |
1 |
1 |
|||
Yes |
2,97 |
0,002 |
|
3,24 |
0,003 |
Diagnosis |
|||||
Surgical |
1 |
1 |
|||
Medical |
2,00 |
0,054 |
|
1,48 |
0,312 |
Sex |
|||||
Female |
1 |
1 |
|||
Male |
0,93 |
0,847 |
|
1,53 |
0,276 |
Benzodiazepines |
|||||
No |
1 |
1 |
|||
Yes |
1,89 |
0,794 |
|
1,70 |
0,475 |
Age |
1,003 |
0,770 |
|
0,999 |
0,930 |
Apache II |
1,056 |
0,024 |
|
1,057 |
0,046 |
*HRaj = Hazard Ratio adjusted |
- Table 2 APACHE II and age are binary variables, but this is not stated in the method.
Response: Thanks for the commentaries. We revised the manuscript and made corrections in the text
- Line 167 The authors seem to be trying to identify the factors that make SSD to delirium transition as an outcome. This does not appear to be consistent with the purpose of this study.
- Response: Thanks for the commentaries. We made changes to better describe the main objectives. Once the SSD to delirium transition is associated with negative outcomes (increase in LOS), we try to identify the main factor associated with worse mental status. As we discussed, SSD can be the first sign of a mental dysfunction or an underlying disease in critically ill patients. Despite an unclear benefit in the treatment of SSD, we believe that SSD monitoring is important for identifying patients at risk for delirium and for improving complementary measures such as sleep control or pharmacological review.
- Line 180 The authors might want to use “associated with”, instead of “impact”.
Response: Thanks for the commentaries. We revised the manuscript, and we made these corrections in the text
- Line 190 Where Table 3 is going to ?
Response: Thanks for pointing this out. The correct is table 2. We corrected the text.
- We do not think it is necessary to include OR=1 for reference in the multivariate analysis table.
Response: Thanks for pointing this out. We changed the table and excluded the term OR=1
Reviewer 3 Report
Dear Authors,
Many thanks for the opportunity to read your work.
The introduction develops the topic rightly.
Methods are clear, and results are well reported.
Discussion
You should expand better on the importance of sub-syndromal delirium and its impact on the ICU.
Limitations: there are further tools to detect delirium in ICU, such as EEG, please discuss this point and limitations
Conclusion: correct
Author Response
We want to thank you for having considered our paper for publication in JCM. We answered point-by-point the comments and highlighted all changes in the manuscript (in red font). We believe the manuscript is now much improved thanks to the reviewers’.
Discussion
You should expand better on the importance of sub-syndromal delirium and its impact on the ICU.
Response: The reviewer is right to point this out. We revised the manuscript and especially the discussion section. We included new comments on the introduction and discussion to highlight the importance and relevance of the SSD in the ICU.
Limitations: there are further tools to detect delirium in ICU, such as EEG, please discuss this point and limitations
Response: Thanks for the commentaries. The use of CAM-ICU to delirium detections had several limitations. In fact, patients heavily sedated or in coma can’t be evaluated and many hypoactive delirium was certainly missed. The use of EEG can increase delirium detection but changes in EEG waveforms to delirium detection need to be better studied and described. Unfortunately, we did not have continuous monitoring of EKG disponible. We discussed this as a limitation of the study.
Round 2
Reviewer 2 Report
I thought the authors had appropriately resolved the issues in response to my comments.